# A Zero-Valent Sulfur Transporter Helps Podophyllotoxin Uptake into Bacterial Cells in the Presence of CTAB

**DOI:** 10.3390/antiox13010027

**Published:** 2023-12-22

**Authors:** Honglei Liu, Huiyuan Yu, Rui Gao, Fulin Ge, Rui Zhao, Xia Lu, Tianqi Wang, Huaiwei Liu, Chunyu Yang, Yongzhen Xia, Luying Xun

**Affiliations:** 1State Key Laboratory of Microbial Technology, Shandong University, 72 Binhai Road, Qingdao 266237, China; lhl@sdu.edu.cn (H.L.); yhyuan@mail.sdu.edu.cn (H.Y.); gaorui@saas.ac.cn (R.G.); folin_ge@sjtu.edu.cn (F.G.); zhaoray@a.sxmu.edu.cn (R.Z.); dzcdcjyk@dz.shandong.cn (X.L.); wangtq777@sdu.edu.cn (T.W.); liuhuaiwei@sdu.edu.cn (H.L.); ycy21th@sdu.edu.cn (C.Y.); 2Shandong Provincial Key Laboratory of Test Technology on Food Quality and Safety, Institute of Quality Standard and Testing Technology for Agro-Products, Shandong Academy of Agricultural Sciences, Jinan 250100, China; 3School of Molecular Biosciences, Washington State University, Pullman, WA 99164-7520, USA

**Keywords:** podophyllotoxin, redox oxidative stress, sulfane sulfur, zero-valent sulfur, YedE1E2

## Abstract

Podophyllotoxin (PTOX) is naturally produced by the plant Podophyllum species. Some of its derivatives are anticancer drugs, which are produced mainly by using chemical semi-synthesis methods. Recombinant bacteria have great potential in large-scale production of the derivatives of PTOX. In addition to introducing the correct enzymes, the transportation of PTOX into the cells is an important factor, which limits its modification in the bacteria. Here, we improved the cellular uptake of PTOX into *Escherichia coli* with the help of the zero-valent sulfur transporter YedE1E2 in the presence of cetyltrimethylammonium bromide (CTAB). CTAB promoted the uptake of PTOX, but induced the production of reactive oxygen species. A protein complex (YedE1E2) of YedE1 and YedE2 enabled *E. coli* cells to resist CTAB by reducing reactive oxygen species, and YedE1E2 was a hypothetical transporter. Further investigation showed that YedE1E2 facilitated the uptake of extracellular zero-valent sulfur across the cytoplasmic membrane and the formation of glutathione persulfide (GSSH) inside the cells. The increased GSSH minimized oxidative stress. Our results indicate that YedE1E2 is a zero-valent sulfur transporter and it also facilitates CTAB-assisted uptake of PTOX by recombinant bacteria.

## 1. Introduction

Podophyllotoxin (PTOX) is an aryltetralin-type lignan produced by certain plants in the genus of *Podophyllum*, including *P. hexandrum* (Ex. *Sinopodophyllum hexandrum* (Royle) T. S. Ying) and *P. versipellis* (Ex. *Dysosma versipellis* (Hance) M. Cheng ex Ying) [1]. Several of its derivatives, including etoposide (VP-16) and teniposide (VM-26) [2,3], are well-known antitumor drugs approved by the US Food and Drug Administration for the treatment of several types of cancer; hundreds of its derivatives, such as 4β-NH-(5-Aminoindazole)-podophyllotoxin [4] and triazole/thiadiazole substituted 4′-demethylepipodophyllotoxin [5], are potential drugs that may have different antitumor bioactivities. The derivatives are normally produced by chemical modifications of PTOX extracted from the plants [6], and overharvesting has put some Podophyllum plants at risk of extinction [7].

In recent decades, extensive structural modifications of PTOX have been performed to enhance the potency of tumor selectivity, improve water solubility, or overcome drug resistance [8,9]. Chemical modification often requires a complex synthesis circuit, with poor chiral selectivity and low yields [10]. Biological synthesis differs from chemical synthesis in catalytical stereochemistry and high efficiency [11]. Efforts have been made to identify enzymes catalyzing the formation of different derivatives [12,13], and these enzymes often produce low yields when performed in microorganisms or tobacco leaves. A key limitation is the limited cellular uptake of the natural PTOX.

The treatment of organic solvents or surfactants may increase membrane permeability and stimulate the transportation of substrates into microbial cells [14,15]. Cetyltrimethylammonium bromide (CTAB) is a cationic surfactant that increases cell membrane permeability. At low concentrations, it may increase substrate transportation, as it has been shown to promote the growth of biofilm [16]. At high concentrations, it is toxic [17]. Since CTAB displays tremendous inhibitory effects on osteosarcoma, its usage as a potential therapeutic drug is under exploration [18]. CTAB is also toxic to *Escherichia coli*, as its treatment of *E. coli* cells increases the production of superoxide and hydrogen peroxide [19]. Thus, CTAB may be used to improve the permeability of bacterial membranes for PTOX uptake when cell viability is maintained.

Sulfane sulfur is a common cellular component [20]. Sulfane sulfur is a collective term of zero-valent sulfur in different forms, including persulfide (RSSH), polysulfide (RSS_n_R), and elemental sulfur in the cells [21]. Sulfane sulfur is generated during the metabolism of sulfur-containing compounds, such as cysteine and methionine, and the oxidation of hydrogen sulfide (H_2_S) by sulfide:quinone oxidoreductase (SQR) [22,23]. Sulfane sulfur is involved in many functions, including redox regulation of protein functions and cell redox homeostasis maintenance [24]. Sulfane sulfur species also act as signal molecules by modifying proteins at Cys residues or reacting with metal centers in proteins [25]. Sulfane sulfur may function as antioxidants as well as oxidants. The S-sulfhydration of Cys residues could protect them from oxidation damage by reactive oxygen species (ROS) [26]. Excessive sulfane sulfur could be toxic to the cells, due to the induced oxidation of cellular thiols [27]. Some bacteria contain persulfide dioxygenase (PDO) that oxidizes sulfane sulfur in the form of glutathione persulfide (GSSH) to sulfite [28], and sulfite further reacts with sulfane sulfur to produce thiosulfate [23]. Thus, it is necessary to maintain homeostasis of intracellular sulfane sulfur.

In this study, a sulfur transporter YedE1E2, consisting of two proteins YedE1 and YedE2, was characterized. It could increase cellular sulfane sulfur levels in bacterial cells, which reduced cellular ROS and improved cell viability when CTAB was used to increase PTOX uptake.

## 2. Materials and Methods

Bacterial strains and culture conditions. The bacterial strains and plasmids used in this study are listed in Table 1. All primers are listed in Appendix A. The *gfp* (accession number JQ064510) was used as control gene in comparison with a functional gene in plasmid constructions. *E. coli* was cultivated in LB medium (10 g/L peptone, 5 g/L yeast powder, 10 g/L NaCl) at 37 °C. *Sinorhizobium meliloti* 1021 and *Cupriavidus pinatubonensis* (*C. pinatubonensis*) JMP134 were cultivated in LB or minimal salt medium (MM) supplemented with 2% glucose [29] at 30 °C. Kanamycin, ampicillin, spectinomycin, isopropyl-β-D-thiogalactopyranoside (IPTG), CTAB, and PTOX were added as required. The culture growth was monitored by using optical density at 600 nm (OD_600nm_). The low-copy plasmids pBBR1MCS2 [30] and pCL1920 [31] and the high-copy plasmid pTrc99a [32] were used to express sulfide:quinone oxidoreductase (SQR), persulfide dioxygenase (PDO), a potential transporter (YedE1E2) for physiological analysis. *CpyedE1E2*, *Cppdo2* and *Cpsqr* were the corresponding genes from *C. pinatubonensis* JMP134, and their encoded proteins were YedE1E2 (protein ID WP_011296423.1 and WP_011296422.1), PDO2 (protein ID WP_011299714.1), and SQR (protein ID WP_011299713.1).

Recombinants and mutants construction. Each vector fragment and the target gene were amplified by PCR and assembled by using the In-Fusion HD Cloning Kit (Clontech, Mountain View, CA, USA). The gene was assembled into the plasmid pBBR1MCS2 by replacing the *lacZα* gene or inserted into the plasmid pTrc99a between the *sacI* and *xbaI* restriction sites. For any construction with the plasmid pCL1920, the process was illustrated in Appendix A. The recombinant plasmids were transformed into *E. coli* BL21 by heat shock. The site-directed mutations of CpYedE1 C104S and CpYedE2 C101S were generated by using a modified QuikChange method [33].

Preparation of crude PDO and its usage in zero-valent sulfur consumption. *E. coli* (PDO2) recombinant cells were cultivated and induced as above. Cells were harvested by centrifugation, resuspended in 50 mM Tris-HCl buffer (pH 7.4) to OD_600nm_ of 5, and disrupted by a Pressure Cell Homogenizer (Stansted Fluid Power Ltd., Harlow, Essex, UK) at 4 °C. The lysate was centrifuged at 12,000× *g* for 10 min to remove cell debris. The supernatant containing soluble PDO2 and GSH was mixed with SQR and YedE1E2-expressing cells at OD_600nm_ = 2, 1 mM sodium hydrogen sulfide (NaHS) was added to the suspension to initiate the sulfide oxidation by SQR and PDO2 at room temperature. The remaining sulfane sulfur in the suspension was analyzed using the KCN method as previously described [23].

PDO purification. *E. coli* (PDO2) recombinant cells were induced, harvested, and disrupted in 20 mM Tris-HCl with 0.5 M NaCl and 20 mM imidazole (pH 8.0). The lysate was centrifuged at 12,000× *g* for 10 min to remove cell debris. PDO was purified by using nickel-nitrilotriacetate (NiNTA) agarose (Qiagen, Shanghai, China) as instructed.

Membrane extraction. *E. coli* (YedE1E2) and *E. coli* BL21 cells were disrupted and centrifuged at 12,000× *g* for 10 min to remove cell debris. The supernatant was centrifuged at 160,000× *g* and 4 °C for 1 h to precipitate the membrane fraction. The pellets were resuspended in 50 mM Tris-HCl buffer (pH 7.4) by stirring with a glass rod.

Preparation of SQR-generated-zero-valent sulfur, chemically synthesized polysulfides, and GSSH. SQR-generated-zero-valent sulfur was prepared by adding 2–4 mM NaHS to *E. coli* (SQR) cells for complete reaction till no sulfide can be detected. The cells were harvested, resuspended, and disrupted by a Pressure Cell Homogenizer at 4 °C and centrifuged at 12,000× *g* for 10 min to remove cell debris. Concentrations of SQR-generated-zero-valent sulfur in the supernatant were measured by the KCN method. Excessive elemental sulfur was dissolved in acetone to prepare saturated sulfur in acetone solution (17–20 mM). GSH was dissolved in 50 mM potassium phosphate buffer (pH 7.4) to 17 mM and mixed with the same volume of saturated elemental sulfur in acetone to prepare 8.5 mM GSSH. Chemically synthesized polysulfides were prepared according to Kamyshny [34]. Briefly, sulfur powder (128 mg) and NaHS (224 mg) were added to 80 mL of anoxic distilled water under argon gas and sealed in a bottle. The bottle was incubated at 37 °C till the sulfur was completely dissolved. The pH was adjusted to 9.3 with 6 M HCl, and anoxic distilled water was added to the solution to a final volume of 100 mL to generate a stock solution of 40 mM chemically synthesized polysulfides.

Whole-cell analysis. *E. coli* with plasmids containing gfp or the testing genes were cultivated at 37 °C with shaking, induced with 0.4 mM IPTG when OD_600nm_ reached approximately 0.6, and further cultivated at 30 °C to OD_600nm_ of 3. The cells were harvested by centrifugation (5000× *g*, 6 min), washed, and resuspended in 50 mM Tris-HCl buffer (pH 7.4) to 2 at OD_600nm_. Approximately 0.5 mM or 1 mM of NaHS, polysulfide, or GSSH was added to the cell suspensions to initiate the reaction. The cell suspensions were incubated at 30 °C, 100 rpm, and the analysis were performed after different time periods.

The KCN method used for the detection of sulfane sulfur in the suspension was the same as previously described; intracellular sulfane sulfur, thiosulfate, and sulfite were derivatized with monobromobimane (mBBr) and detected by using HPLC [22,23,28].

PTOX analysis. PTOX (Shanxi Huisheng Medicament Technology Company, Xi’an, China) was dissolved in dimethyl sulfoxide (DMSO) to 100 mM before being used as the stock solution. Totally 5 mL of PTOX reaction or cultivation solution was diluted with acetonitrile to 10 mL and centrifuged at 12,500× *g* for 30 min. The supernatant was filtered with a 0.45 μm-micropore filter followed by HPLC analysis.

HPLC analysis was performed on a Shimadzu LC-20AT HPLC system with an ultraviolet detector and Akasil C18 column (5 μm, 4.6 mm × 250 mm). The column was eluted with a gradient of solution A (ddH_2_O) and solution B (acetonitrile) from 7.5% B to 52.5% B in 1 min, 55% B for 15 min, and 100% B for 15 min at a flow rate of 0.8 mL/min. The detection wavelength was 254 nm.

Cellular ROS analysis. Cellular ROS was analyzed by using a Bacterial ROS Analysis Kit (Beijing Baiaolaibo Technology Co., Ltd., Beijing, China).

Analysis of YedEs in sequenced bacterial genomes. A microbial genomic sequence set from NCBI (updated to 18 January 2019) was downloaded, and CpYedE1 (WP_011296423.1), CpYedE2 (WP_011296422.1) and EcYedE3 (protein ID CAQ32419) were used as query to search the database with the standalone BLASTP algorithm, using conventional criteria (E value of ≤1 × 10^−5^, coverage of ≥50%, and identity of ≥50%). YedE candidates from a total of 8671 bacterial genomes were obtained. The candidates were combined with the seed YedEs for phylogenetic tree analysis using ClustalW for alignment and MEGA 7.0 for neighbor-joining tree building, with pairwise deletion, p-distance distribution, and bootstrap analysis of 1000 repeats as parameters. The candidates that were in the same clade as the seed YedEs were picked for further analysis. The identified YedE sequences and seed sequences were separately grouped into unique groups by using the CD-HIT program, with an identity of ≥50% as the threshold.

## 3. Results

### 3.1. CTAB Improved PTOX Uptake in E. coli

The transport of PTOX into *E. coli* cells was the first step to modify it by intracellular enzymes. When 200 μM PTOX was added to *E. coli* cultures at OD_600nm_ of 2.5 and further cultivated for 5 h, the cells displayed almost no uptake of PTOX, but the addition of 200 μM CTAB significantly enhanced the uptake (Figure 1). Unfortunately, nearly half of *E. coli* BL21 cells were lysed after 5 h. At the same time, we were investigating a potential transporter encoded by *yedE1* and *yedE2* from *C. pinatubonensis* JMP134, and whether the transporter helped PTOX uptake was tested. We used *E. coli* (YedE1E2) cells that overexpressed *yedE1* and *yedE2* to test PTOX uptake with or without CTAB. Although *E. coli* (YedE1E2) cells did not take up PTOX, the presence of YedE1E2 stabilized the cells with CTAB. The cellular concentration of PTOX in *E. coli* (YedE1E2) cells reached 369–885 μM (Figure 1). These results indicate that YedE1E2 helps the cells to resistant CTAB, but not directly stimulates PTOX uptake.

### 3.2. Sequence Analysis Indicates YedE and Yed1E2 Are Membrane Proteins That form a Potential Transporter

The YedE genes are often located next to known sulfur metabolism genes present in bacterial genomic DNA. In *C. pinatubonensis* JMP 134, the *yedE1* and *yedE2* genes are located next to an ArsR/SmtB family transcription factor, and close to a tRNA 2-thiocytidine biosynthesis protein TtcA that catalyzes the thiolation of tRNA. In *Agrobacterium tumefaciens*, *yedE1E2* are upstream of the *blh* gene encoding a PDO, that catalyzes the oxidation of GSSH to sulfite and GSH. STRING analysis of *yedE1E2* revealed the co-occurrence of them with sulfur metabolism genes in its vicinity, including genes coding for rhodanese, PDO, SQR, TauE (sulfite transporter), and CysN (sulfate adenylyl transferase subunit), in bacteria and fungi, implying that these proteins may be involved in sulfur metabolism (Appendix A). In *Serratia* sp. strain ATCC 39006, two membrane proteins PmpA and PmpB have been proposed to transport sulfur-containing molecules [35]. CpYedE1 and CpYedE2 have 59% sequence identities to PmpA and PmpB, respectively, and they all belong to the COG2391 family of putative integral membrane proteins.

In *C. pinatubonensis* JMP 134, CpYedE1(WP_011296423.1) and CpYedE2 (WP_011296422.1) are small proteins of 14.3 kD and 14.6 kD, respectively. In *S. meliloti* strain 1021, SmYedE1(AAK65228.1) and SmYedE2 (AAK65227.1) were 13.6 kD and 15 kD proteins, respectively. CpYedE1 and CpYedE2 were homologous with 27% of sequence identity. CpYedE1 and CpYedE2 shared 53% and 46% identities with SmYedE1 and SmYedE2, respectively. In *E. coli* BL21 (DE3), EcYedE3, a 39 kD protein, was likely a fusion of YedE1 and YedE2 that shared 30% sequence identity. The TMHMM protein transmembrane analysis showed that YedE1 and YedE2 each contained four transmembrane helixes, and EcYedE3 had nine transmembrane helixes. YedE3 mediates thiosulfate uptake in *E. coli, Spirochaeta thermophila, and Metallosphaera cuprina* [36,37,38]. Thus, YedE1 and YdeE2 were likely forming a transporter (YedE1E2). The physiological function of YedE1E2 was unclear, but the results shown in Figure 1 suggest that the two membrane proteins may resist CTAB toxicity.

### 3.3. E. coli (YedE1E2) Was More Tolerant to CTAB Than E. coli BL21

*E. coli* (YedE1E2) and *E. coli* BL21 in LB medium with 200 μM PTOX and varying concentrations of CTAB from 0 to 600 μM were cultivated. The growth and CTAB tolerance of each strain were detected. CTAB was more inhibitory to *E. coli* BL21 than to *E. coli* (YedE1E2) (Figure 2A,B). After 20 h cultivation, 600 μM CTAB completely inhibited the growth of *E. coli* BL21, and even 200 μM CTAB inhibited its growth for the first 10 h (Figure 2A). However, *E. coli* (YedE1E2) cells started to grow after 15 h in the presence of 600 μM CTAB (Figure 2B). When both *E. coli* BL21 and *E. coli* (YedE1E2) started to grow in the presence of CTAB, the growth rates were similar to those without CTAB (Figure 2A,B). The delayed growth is likely due to the toxicity of CTAB, and the late start of the growth could be due to the accumulation of mutations or the activation of resistant genes, which should be further investigated. On the agar plate, 400 μM CTAB was partially inhibitory to *E. coli* (YedE1E2) but completely inhibitory to *E. coli* BL21 (Figure 2C).

### 3.4. YedE1E2 Expressing Cells Reduced Cellular ROS by Regulating Zero-Valent Sulfur Content in the Cells

CTAB has cytotoxicity and may cause ROS elevation in cells. During cultivation of *E. coli* BL21 and *E. coli* (YedE1E2), 200 μM CTAB was added when the culture reached OD_600nm_ = 2.5 and further cultivated for 5 h before cells were taken for ROS analysis. CTAB increased cellular ROS in *E. coli* BL21, but *E. coli* (YedE1E2) contained less ROS than *E. coli* BL21 (Figure 3A). When H_2_O_2_ was used as the positive control, it also increased ROS less in *E. coli* (YedE1E2) than in *E. coli* BL21 (Figure 3A). In the presence of 200 μM CTAB, *E. coli* (YedE1E2) contained less sulfane sulfur, but produced more thiosulfate (Figure 3B,C). We speculated that YedE1E2 facilitated the reaction of sulfane sulfur with GSH in the cytoplasm to produce reactive GSSH, which reacted with ROS. The reaction consumed sulfane sulfur, produced thiosulfate, and lowered ROS. The speculation also holds up in the absence of CTAB, as *E. coli* (YedE1E2) contained less ROS than *E. coli* BL21 (Figure 3A).

### 3.5. YedE1E2 Sped up the Formation of GSSH

To detect whether YedE1E2 sped up the formation of GSSH, PDO was used to oxidize GSSH, which was formed between GSH and acetone-dissolved elemental sulfur. The reaction mixture also contained membrane fractions of *E. coli* (YedE1E2) or *E. coli* BL21. The reaction system containing the *E. coli* (YedE1E2) membrane fractions produced more thiosulfate than the system containing the *E. coli* BL21 membrane fractions (Figure 4). Sulfite was not detected in the first 3 min, as PDO oxidizes GSSH to GSH and sulfite, and the latter spontaneously reacts with excessive zero-valent sulfur to produce thiosulfate [28]. The results suggest that YedE1E2 facilitated the reaction of GSH and zero-valent sulfur to produce GSSH.

To further test whether YedE1E2 facilitated the transport of zero-valent sulfur across the cytoplasmic membrane, the gene *pdo2* from *C. pinatubonensis* JMP134 was cloned into *E. coli* BL21 and *E. coli* (YedE1E2) to produce *E. coli* (PDO2) and *E. coli* (PDO2/YedE1E2). The two strains were used to oxidize three types of zero-valent sulfur: SQR-generated-zero-valent sulfur, chemically synthesized polysulfides, and acetone-dissolved elemental sulfur. The sulfur preparations were added at 1 mM to 10 mL of the cell suspension at OD_600nm_ of 2 in 50 mM, pH 7.4 Tris-HCl buffer. *E. coli* (PDO2/YedE1E2) oxidized them much faster than *E. coli* (PDO2) did (Figure 5A–C). Thus, YedE1E2 stimulates the uptake of polysulfides and elemental sulfur. Thiosulfate was the main product when chemically synthesized polysulfides were oxidized by both strains, and *E. coli* (PDO2/YedE1E2) produced more thiosulfate than *E. coli* (PDO2) did (Figure 5D). Sulfite was low, and less than 20 μM was detected in all the samples. The conversion of extracellular zero-valent sulfur to intracellular GSSH is also a type of transportation, which is facilitated by YedE1E2.

### 3.6. YedE1E2 Stimulated the Import but Not the Export of Zero-Valent Sulfur

*E. coli* with cloned SQR readily oxidizes added sulfide to zero-valent sulfur in the cytoplasm [21]. When the sulfur-containing cells are mixed with PDO or *E. coli* cells with cloned PDO2 in 50 mM Tris-HCl buffer (pH 7.4), zero-valent sulfur is rapidly oxidized to thiosulfate [21]. We tested whether YedE1E2 helped the export and import of zero-valent sulfur. Zero-valent sulfur was produced in the cytoplasm of *E. coli* (SQR) and *E. coli* (SQR/YedE1E2). The sulfur-containing cells were mixed with crude PDO, *E. coli* (PDO2), or *E. coli* (PDO2/YedE1E2). YedE1E2 in *E. coli* (SQR/YedE1E2) cells with zero-valent sulfur did not speed up sulfur oxidation by PDO (Figure 6A). Further, YedE1E2 in *E. coli* (SQR/YedE1E2) cells with zero-valent sulfur did not speed up sulfur oxidation by *E. coli* (PDO2) cells. However, YedE1E2 in *E. coli* (PDO2/YedE1E2) cells oxidized sulfur much faster than *E. coli* (PDO2) did (Figure 6). The results suggest that YedE1E2 speeds up the import but not the export of zero-valent sulfur.

### 3.7. YedE1E2 Homologues Were Likely Involved in the Uptake of Zero-Valent Sulfur but the Homologous YedE3 Was Not

We further tested *S. meliloti* 1021 YedE1 (Protein ID AAK65228.1) and YedE2 (Protein ID AAK65227.1) (SmYedE1E2), *E. coli* YedE3 (Protein ID CAQ32419; EcYedE3) and *C. pinatubonensis* JMP134 YedE3 (Protein ID AAZ60179.1; CpYedE3) for sulfur uptake activities. The genes encoding YedEs were cloned with PDO in *E. coli. E. coli* (PDO2/SmYedE1E2) oxidized chemically prepared polysulfide as fast as *E. coli* (PDO2/CpYedE1E2), while *E. coli* (PDO2/EcYedE3), *E. coli* (PDO2/CpYedE3) and *E. coli* (PDO2) oxidized chemically prepared polysulfide at similar slow rates. *E. coli* (PDO2/CpYedE1) and *E. coli* (PDO2/CpYedE2) also did not show any increased rates of the oxidation (Figure 7). Thus, YedE1 and YedE2 together speed up the uptake of zero-valent sulfur, but YedE1 alone, YedE2 alone, and YedE3 do not.

### 3.8. The Conserved Cys Residues Are Essential for YedE1E2 Activities

When CpYedE1 and CpYedE2 were aligned with other YedE1 and YedE2 proteins, both proteins had a conserved Cys residue, which are CpYedE1 Cys104 and CpYedE2 Cys101. The TMHMM protein transmembrane analysis indicated that the CpYedE2 Cys101 was in the middle of a transmembrane α-helix and CpYedE1 Cys104 was at the end of a transmembrane α-helix facing the cytoplasm. Mutagenesis of Cys104 in CpYedE1 or Cys101 in CpYedE2 to a Ser residue abolished the uptake activity (Figure 8A) and growth resistance to CTAB (Figure 8B). When exposed to CTAB, the cellular sulfane sulfur and ROS in *E. coli* (YedE1E2 C101S) and *E. coli* (YedE1E2 C104S) were similar to those in *E. coli* BL21(DE3) (Figure 8C,D), indicating the two conserved Cys residues in YedE1E2 are important for sulfur uptake.

### 3.9. Distribution of YedEs in Bacteria

We searched *yedE* genes in the 8671 genomes from GenBank. A total of 847 YedE proteins were identified in 550 bacterial genomes, which were clustered into YedE1, YedE2, and YedE3 subgroups on the phylogenetic tree (Figure 9). We identified 359 YedE1s in 342 strains, 327 YedE2s in 307 strains, and 161 YedE3s in 159 strains (Table 2). The majority of strains containing YedE1 or YedE2 were found in alpha-, beta-, gamma- or deltaproteobacteria, with only six of YedE1- and two of YedE2-containing strains belonging to other phyla. Most YedE3s exist in the Proteobacteria; however, approximately 30 Gram-positive bacilli contain YedE3.

Among the 212 strains containing both YedE1 and YedE2, most have one pair of YedE1E2, and 7 strains contain two pairs of YedE1E2, with the majority of these *yedE1* and *yedE2* genes being neighbors on the chromosome. Approximately 20% of YedE3 were found in the same strain containing YedE1 and YedE2, such as in *C. pinatubonensis* JMP134 and *S. meliloti* 1021.

## 4. Discussion

Biosynthesis of secondary metabolites derived from plants in prokaryotic organisms is an ongoing trend in the drug industry [39,40,41,42]. The secondary metabolites are often in small amounts in the plant cells, which often puts them in short supply and requires large amounts of capital investment on their extraction. Chemical and biological syntheses are important alternatives to obtain the metabolites and their derivatives, which are critical drugs. Compared to chemical synthesis that may be cumbersome with lots of protection and deprotection steps, biosynthesis could be a good choice to produce the drugs due to the strict stereo- and regioselectivity, mild production conditions, and easy operation [43,44]. In vitro micro-propagated plant cultures, cell suspensions, hairy root cultures, genetically modified higher transgenic plants, and recombinant microbes are often applied to produce these metabolites [43]. In order to obtain high yields, it is always promising to use effective molecular tools to accelerate the reconstitution of plant biosynthetic pathways in microorganisms. However, the synthesis of some drugs is complicated with multi-step enzymatic reactions [45]. Thus, the biosynthesis using precursors is of great import in the production of drugs. Microbial production is widely used because of several advantages, such as easy genetic operation, simple cultivation, high yield, and low cost. The biosynthesis of some intermediates or derivatives of PTOX are carried out in bacteria [44,46,47]; however, the complexity of the substrates often makes it hard to transport them into the cells. CTAB, a surfactant, breaks the integrity of bacterial cytomembrane and promotes the uptake of prodrugs. Two transmembrane proteins, YedE1 and YedE2, together greatly increased the absorbance of PTOX by the cells in the presence of CTAB, this offers a good prospect for the use of bacteria cells to produce drug derivatives from prodrugs (Figure 1). Further investigations are necessary to demonstrate whether YedE1E2 can be used to promote CTAB-enhanced uptake of prodrugs like PTOX for the production of their derivatives in recombinant microorganisms and whether the production is economically feasible.

CTAB is toxic to the cells, and it induces superoxide stress in *E. coli* cells, leading to a decrease in cell viability [19]. Although CTAB does not change the viability of bacteria in biofilms [16], planktonically growing bacteria do not have stress resistance features, which is in agreement with our observation (Figure 2A,B). However, *E. coli* (YedE1E2) cells exhibited much better adaptation to CTAB (Figure 2C). One possible explanation is that YedE1E2 directly fortifies the membrane integrity caused by CTAB. If so, *E. coli* (YedE1E2) should not be able to uptake more PTOX in the presence of CTAB, but this is not the case (Figure 1), suggesting the cell membrane was still compromised by CTAB. Thus, the expression of YedE1E2 may have other impacts on the growth of *E. coli* cells. CTAB induces the generation of superoxide and hydrogen peroxide in *E. coli* [19], and the overproduction of ROS is a major cause of cell death through damaging critical cellular constituents, altering redox homeostasis, and initiating a variety of signaling responses [48,49,50]. Our results suggest that YedE1E2 reduced ROS induced by CTAB in *E. coli* (YedE1E2) cells (Figure 3A).

YedE1E2 consists of two YedE (or YeeE) transmembrane proteins that belong to the sulfur transport superfamily. We searched the appearance of PDO in YedE1E2-containing bacteria and found that 177 of the 212 (approximately 83.5%) yedE1E2-containing bacteria have *pdo* genes in their genomes. Walsh et al. attribute the *yedE1-yedE2-pdo2* operon to sulfur detoxification in *Acinetobacter baumannii* [51]. Since PDO oxidizes GSSH to sulfite [28], the coexistence of *yedE* genes with *pdo* suggests that the uptake of zero-valent sulfur is associated with their consumption inside the cells. When coupled to PDO inside the cells, YedE1E2 actively transported zero-valent sulfur into the cells and produced reactive small thiol persulfide, such as GSSH (Figure 5). The sulfur transfer is directional, as YedE1E2 does not help the transport of zero-valent sulfur out of the cells (Figure 6). The directional transportation of zero-valent sulfur is likely due to the localization of the two conserved Cys residues YedE1 Cys104 and YedE2 Cys101, which are essential for the activities of YedE1E2 (Figure 8).

The mechanism of how YedE1E2 reduces oxidative stress to *E. coli* cells in the presence of CTAB is likely due to its ability to produce GSSH. Three lines of evidence support this hypothesis. First, the membrane fractions containing YedE1E2 facilitate GSSH formation from GSH and zero-valent sulfur (Figure 4). The dominant species of zero-valent sulfur is S_8_, which has very low solubility in aqueous solutions and is less reactive due to the low solubility. On the other hand, GSSH is soluble and readily reacts with ROS to protect cells [52]. Second, *E. coli* (YedE1E2) contains less cellular sulfane sulfur and more thiosulfate when exposed to CTAB (Figure 3B,C). Here, we suspect that YedE1E2 facilitated the reaction of less reactive sulfane sulfur with GSH to produce GSSH, GSSH then reacts with ROS, producing thiosulfate [52]. Third, the transport and anti-ROS activities are likely due to the presence of the conserved Cys residues in YedE1 (Cys104) and YedE2 (Cys101) (Figure 8). In structure models, Cys101 of YedE2 is in the middle of the membrane and Cys104 of YedE1 is on the cytoplasmic side of the membrane. Mutations in either Cys101 or Cys104 abolished the activity of YedE1E2 (Figure 8A). The two Cys residues may facilitate the formation of GSSH. Sulfur modification of protein Cys thiols is common inside cells to produce persulfide or even polysulfide -SSnH (n ≥ 1) [53,54]. As a membrane protein, YedE1E2 could also react with sulfane sulfur through its Cys residues to produce persulfide -SSH, which is then transferred to nearby acceptors like GSH to be released as GSSH (Figure 4). This type of process has been reported with a rhodanese, which receives sulfane sulfur with the active site Cys residue and then passes it to GSH [55]. GSSH is reactive and is a potent antioxidant inside cells [56]. Alternatively, YedE1E2 Cys persulfide may also directly react with ROS, which also leads to reduced cellular sulfane sulfur and increased thiosulfate in *E. coli* (YedE1E2) (Figure 3).

## 5. Conclusions

Our results show that the transporter YedE1E2 is able to transport zero-valent sulfur into bacterial cells and produce GSSH inside the cytoplasm. The uptake is enhanced if GSSH is consumed by PDO. Further, YedE1E2 increases cellular sulfane sulfur even without extracellular zero-valent sulfur, which helps the cells to deal with oxidative stress and tolerate CTAB. The new transporter may be used to enhance the oxidation of extracellular zero-valent sulfur and the uptake of PTOX by bacteria.

## 6. Patents

Chinese Patent (No. 202310719025.6) results from the work reported in this manuscript.

## Figures and Tables

**Figure 1 antioxidants-13-00027-f001:**
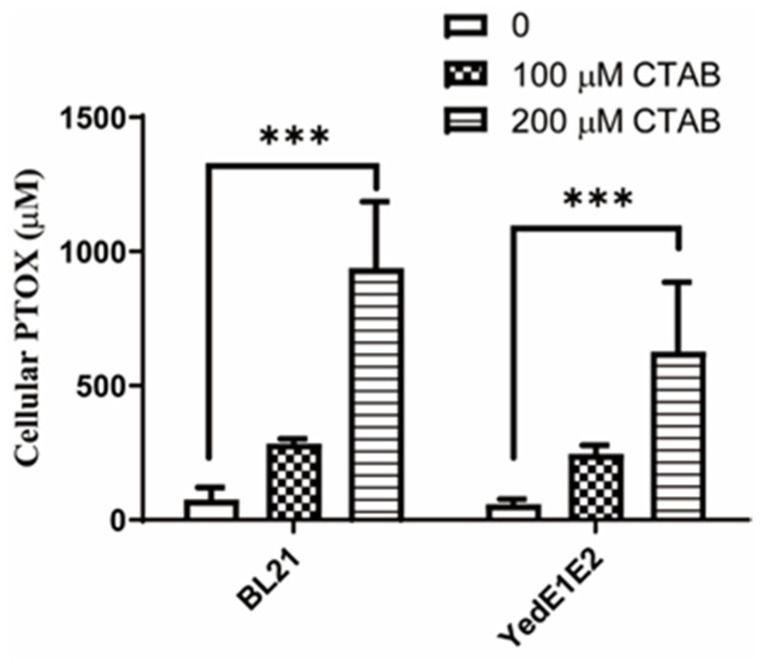
CTAB stimulated the cellular uptake of PTOX. *E. coli* BL21 and *E. coli* (YedE1E2) were cultivated in LB medium at 37 °C, 200 rpm, and YedE1E2 production was induced with IPTG. CTAB of 0 to 200 μM and PTOX of 200 μM were added when OD_600nm_ reached 2.5 and further cultivated for 5 h, cellular PTOX concentration was detected by using the HPLC method after cell lysis by sonication. Data are the averages of three measures with standard deviation. *p*-values were calculated by One-way ANOVA, and asterisks indicate statistically significant differences (***, *p* < 0.0001).

**Figure 2 antioxidants-13-00027-f002:**
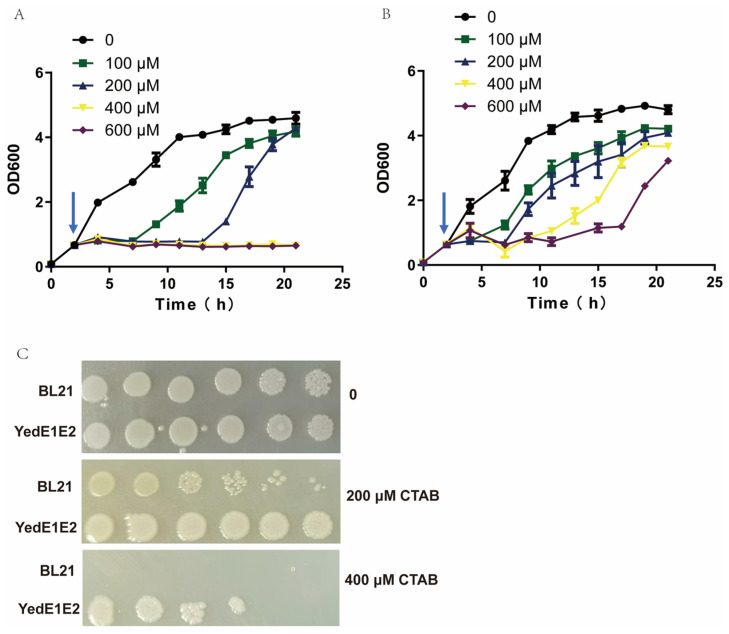
*E. coli* cells with YedE1E2 were resistant to CTAB. *E. coli* BL21 (**A**) and *E. coli* (YedE1E2) (**B**) were inoculated with 1% overnight cultures and cultivated in LB medium at 37 °C for 2 h to OD_600nm_ of around 0.6, and then 400 μM IPTG and different concentrations of CTAB were added. The cultures were further cultivated and monitored for 19 h. Arrows in (**A**,**B**) marked the time when IPTG and CTAB were added. (**C**) YedE1E2 increased bacterial tolerance to CTAB on LB agar plates. *E. coli* BL21 was the control, containing the same plasmid with a *gfp* gene.

**Figure 3 antioxidants-13-00027-f003:**
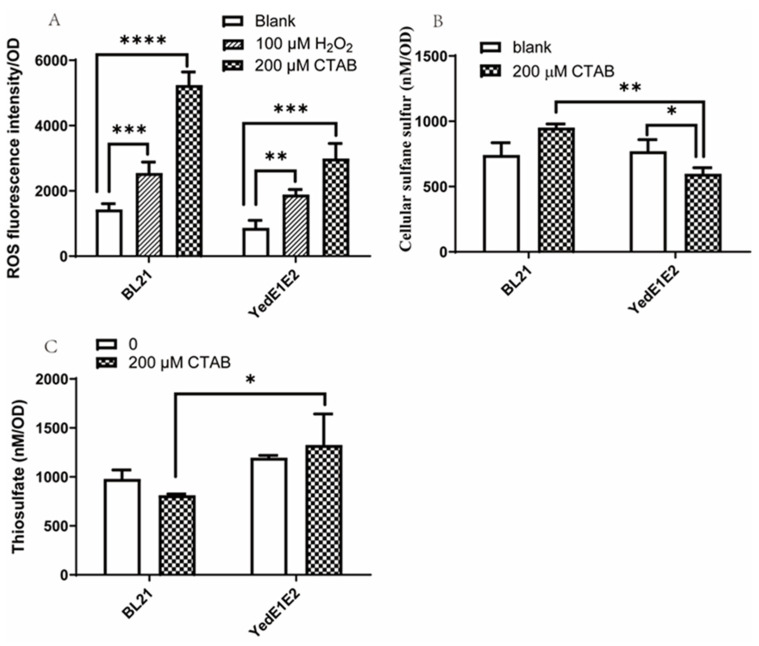
*E. coli* (YedE1E2) cells reduced cellular ROS by consuming cellular sulfane sulfur. *E. coli* BL21 and *E. coli* (YedE1E2) were cultivated in LB medium at 37 °C, 200 rpm to OD_600nm_ = 0.6 and induced with 0.4 mM IPTG. When OD_600nm_ = 2.5, 500 μM PTOX was added with either 200 μM CTAB or 100 μM H_2_O_2_ and cultivated for 5 h. Cells were harvested, washed, and resuspended for ROS analysis. (**A**) Cellular ROS generation; (**B**) cellular sulfane sulfur; (**C**) thiosulfate accumulation. Data are the averages of three measures with standard deviation. *p*-values were calculated by One-way ANOVA, and asterisks indicate statistically significant differences (* *p* < 0.05, ** *p* < 0.001, *** *p* < 0.0001, **** *p* < 0.00001).

**Figure 4 antioxidants-13-00027-f004:**
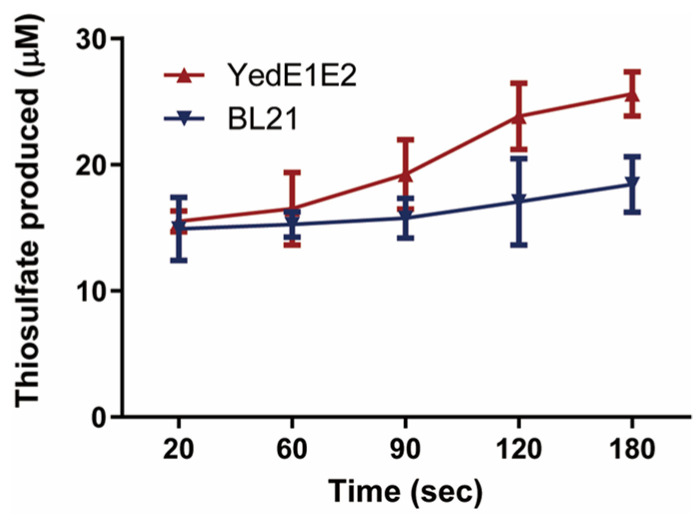
*E. coli* membrane fractions containing YedE1E2 accelerated the formation of GSSH from GSH and zero-valent sulfur. The reaction mixture contained 20 μg of purified CpPDO2, 1 mM GSH, and 1 mM acetone-dissolved zero-valent sulfur in 1 mL of 50 mM Tris-HCl buffer (pH 7.4) with the membrane fractions of either *E. coli* BL21 or *E. coli* (YedE1E2) cells. The produced thiosulfate was detected. Data are the averages of three measures with standard deviation.

**Figure 5 antioxidants-13-00027-f005:**
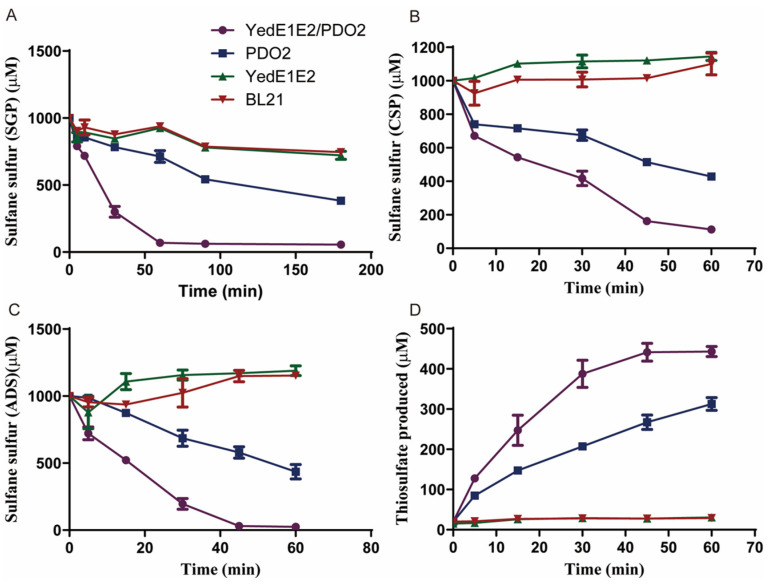
YedE1E2 sped up inorganic zero-valent sulfur consumption by recombinant *E. coli* with PDO. *E. coli* (PDO2) and *E. coli* (PDO2/YedE1E2) cells were resuspended in 50 mM Tris-HCl buffer (pH 7.4) to OD_600nm_ = 2, and the cells were used to oxidize several types of zero-valent sulfur. The following was added to initiation the reaction: (**A**) 1 mM of SQR-generated-zero-valent sulfur (SGS), (**B**) 1 mM chemically synthesized polysulfides (CSP), or (**C**) 1 mM acetone-dissolved elemental sulfur (ADS). (**D**) Thiosulfate generated from the oxidation of CSP was analyzed. The *E. coli* stains contained two plasmids with a tested gene or *gfp* for comparison. Data are the averages of three measures with standard deviation.

**Figure 6 antioxidants-13-00027-f006:**
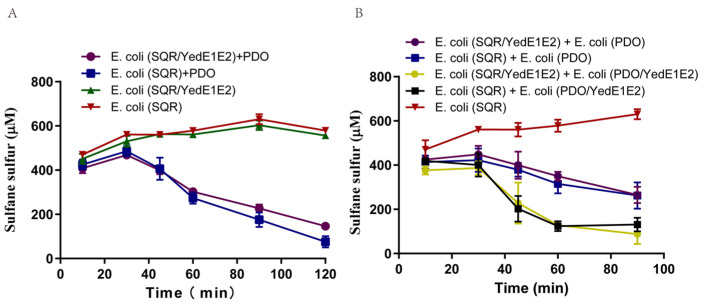
YedE1E2 stimulated the import but not the export of zero-valent sulfur. Cells were resuspended in 50 mM Tris-HCl buffer (pH 7.4) to OD_600nm_ = 2. Intracellular zero-valent sulfur was produced from the oxidation of 1 mM sulfide by *E. coli* (SQR/YedE1E2) and *E. coli* (SQR). (**A**) The cells containing zero-valent sulfur were mixed with crude PDO; (**B**) the cells containing zero-valent sulfur were mixed with *E. coli* (PDO2) or *E. coli* (PDO2/YedE1E2). The *E. coli* stains contained three plasmids with the tested gene or gfp for comparison. Data are the averages of three measures with standard deviation.

**Figure 7 antioxidants-13-00027-f007:**
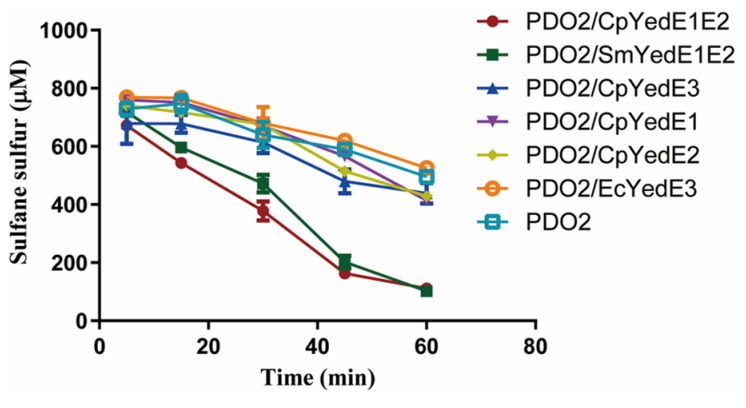
The uptake of zero-valent sulfur by YedE1E2, YedE1, YedE2, and YedE3. *E. coli* cells (with CpPDO2 expressed in pCL1920 and various YedEs expressed in pBBR1MCS2) were resuspended to OD_600nm_ = 2 in 50 mM Tris-HCl buffer (pH 7.4). Chemically synthesized polysulfides were added to a final concentration of 1 mM to start the oxidation. The *E. coli* stains contained two plasmids with a tested gene or *gfp* for comparison. Data are the averages of three measures with standard deviation.

**Figure 8 antioxidants-13-00027-f008:**
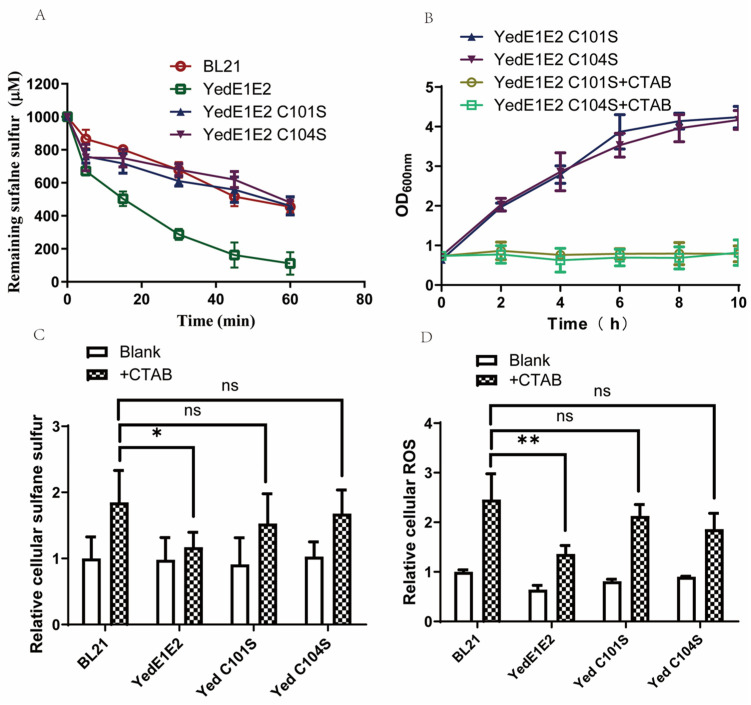
Mutation of Cys101 in YedE2 and Cys104 in YedE1 abolished YedE1E2 activity. (**A**) The oxidation of zero-valent sulfur by *E. coli* (PDO2), *E. coli* (PDO2) with YedE1E2, and its mutants. The *E. coli* cells were resuspended in 50 mM Tris-HCl buffer (pH 7.4) to OD_600nm_ = 2. Chemically synthesized polysulfides were added to a final concentration of 1 mM to start the reaction. (**B**) Growth inhabitation of the mutants. Strains were cultivated in LB medium at 37 °C, 200 rpm to OD_600nm_ = 0.6 and induced by 0.4 mM IPTG, 200 μM CTAB was added, and the optical density was measured for 10 h. (**C**) Relative cellular sulfane sulfur. (**D**) Relative cellular ROS level. Strains were cultivated in LB medium at 37 °C, 200 rpm to OD_600nm_ = 0.6 and induced by 0.4 mM IPTG, CTAB of 200 μM and PTOX of 200 μM were added when OD_600nm_ = 2.5 for 5 h. Cells were harvested, washed, and resuspended for cellular sulfane sulfur and ROS analysis. The *E. coli* stains contained the same plasmid with a tested gene or *gfp* for comparison. Data are the averages of three measures with standard deviation. *p*-values were calculated by One-way ANOVA, and asterisks indicate statistically significant differences (* *p* < 0.05, ** *p* < 0.001, ns, no significant).

**Figure 9 antioxidants-13-00027-f009:**
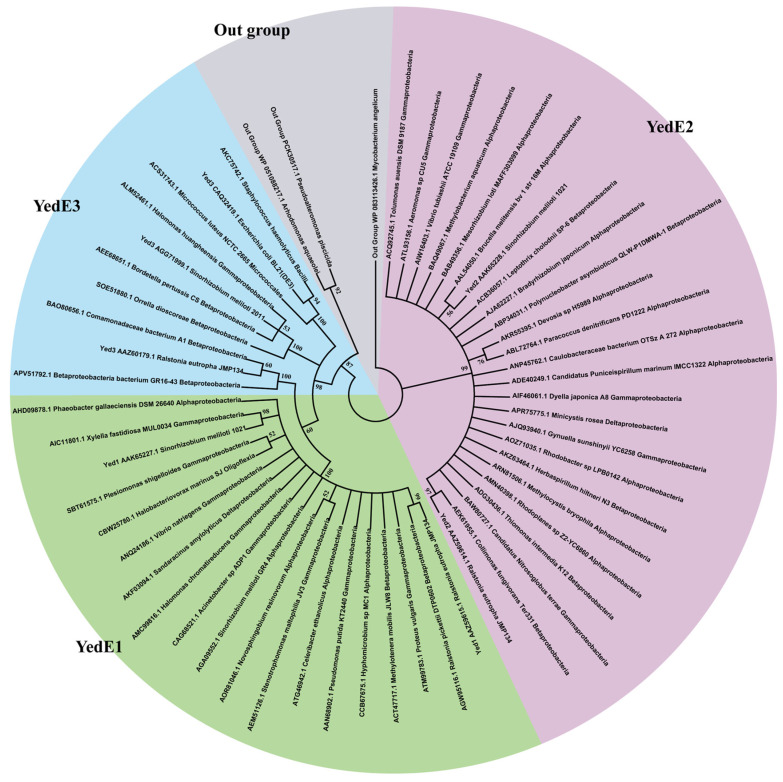
Phylogenetic analysis of YedEs. The tree was generated by using the software MEGA 7. Proteins are listed with their accession numbers and organism origins.

**Table 1 antioxidants-13-00027-t001:** Plasmids and bacterial strains used in this study.

Plasmids and Bacterial Strains	Relevant Characteristic(s)	Source or Reference
Plasmids		
pBBR1MCS2	Km^r^, expression vector	[30]
pCL1920	Spc^r^, expression vector	[31]
pTrc99a	Amp^r^, expression vector	[32]
pBBR2-*gfp*	pBBR1MCS2 that contained *gfp*	This study
pBBR2-*CpyedE1E2*	pBBR1MCS2 that contained *yedE1* and *yedE2* (Reut_A0233, 0232) from *C. pinatubonensis* JMP134	This study
pBBR2-*SmyedE1E2*	pBBR1MCS2 that contained *yedE1* and *yedE2* (Sma1053, 1052) from *S. meliloti* 1021	This study
pBBR2-*CpyedE1*	pBBR1MCS2 that contained *yedE1* from *C. pinatubonensis* JMP134	This study
pBBR2-*CpyedE2*	pBBR1MCS2 that contained *yedE2* from *C. pinatubonensis* JMP134	This study
pBBR2-*CpyedE3*	pBBR1MCS2 that contained *yedE3* gene (Reut_A0799) from *C. pinatubonensis* JMP134	This study
pBBR2-*EcyedE3*	pBBR1MCS2 that contained *yedE3* gene (B21_01902) from *E. Coli* BL21(DE3)	This study
pBBR2-*CpyedE1E2* C101S	pBBR1MCS2 that contained *yedE1 and yedE2* with a C101S mutation in CpYedE2	This study
pBBR2-*CpyedE1E2* C104S	pBBR1MCS2 that contained *yedE1* and *yedE2* with a C104S mutation in CpYedE1	This study
pTrc99a-*CpSqr*	pTrc99a that contained *Cpsqr* gene from *C. pinatubonensis* JMP134	This lab
pTrc99a-*Cppdo2*	pTrc99a that contained *Cppdo2* gene from *C. pinatubonensis* JMP134	This study
pTrc99a-*gfp*	pTrc99a that contained *gfp* gene	This lab
pCL1920-*Cppdo2*	pCL1920 that contained *Cppdo2* gene from *C. pinatubonensis* JMP134	This lab
pCL1920-*gfp*	pCL1920 that contained *gfp* gene	This lab
Strains		
*E. coli* BL21(DE3)	F^−^*ompT hsdS*_B_ (r_B_^−^ m_B_^−^) *gal dcm met* (DE3)	Invitrogen
*E. coli* (CpYedE)	*E. coli* BL21(DE3) with pBBR2-*CpyedE*: *yedE1*, *yedE2*, *yedE1E2,* or *yedE3* from *C. pinatubonensis* JMP134	This study
*E. coli* (SmYedE)	*E. coli* BL21(DE3) with pBBR2-*SmyedE*: *yedE1*, *yedE2*, or *yedE1E2* from *S. meliloti* 1021	This study
*E. coli* (PDO2/CpYedE)	*E. coli* BL21(DE3) with pTc99a-*Cppdo2* or pCL1920-*Cppdo2* (stated specifically) and pBBR2-*CpyedE*: *yedE1*, *yedE2*, *yedE1E2,* or *yedE3* from *C. pinatubonensis* JMP134 BL21(DE3)	This study
*E. coli* (PDO2/SmYedE)	*E. coli* BL21(DE3) with pCL1920-*CpPdo2* and pBBR2-*SmyedE*: *yedE1*, *yedE2*, or *yedE1E2* from *S. meliloti* 1021	This study
*E. coli* (PDO2/EcYedE3)	*E. coli* BL21(DE3) with pCL1920-*Cppdo2* and pBBR2-*EcyedE3*	This study
*E. coli* (PDO2)	*E. coli* BL21 with pTc99a-*Cppdo2* or pCL1920-*Cppdo2* (stated specifically) and pBBR2-*gfp* (or pBBR1MCS2 for ROS analysis)	This study
*E. coli* (SQR/YedE)	*E. coli* BL21(DE3) with pTc99a-*CpSqr* and pBBR2-*yedE*: *yedE1*, *yedE2*, *yedE1E2,* or *yedE3* from *C. pinatubonensis* JMP134	This study
*E. coli* (SQR)	*E. coli* BL21 with pTc99a-*Cpsqr* and pBBR2-*gfp*	This study
*C. pinatubonensis* JMP134	Wild type	Ron L. Crawford, University of Idaho
*S. meliloti 1021*	Wild type	Michael Kahn, Washington State University

**Table 2 antioxidants-13-00027-t002:** Taxonomic distribution of YedE subgroups in the 550 bacterial genomes.

Taxon	No. of YedE Proteins		
	YedE1	YedE2	YedE3
Alphaproteobacteria	71	91	20
Betaproteobacteria	114	108	50
Deltaproteobacteria	13	4	
Gammaproteobacteria	138	102	46
Epsilonproteobacteria			1
Actinomycetales			3
Bacilli			30
Micrococcales			6
Propionibacteriales			2
Rubrobacteria	2	1	1
Nostocales	1		
Oligoflexia	2	1	
Pleurocapsales	1	1	

## Data Availability

Data are contained within this article and Appendix A.

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
