# Peer review of "A Zero-Valent Sulfur Transporter Helps Podophyllotoxin Uptake into Bacterial Cells in the Presence of CTAB"

_antioxidants, 2023, doi:10.3390/antiox13010027_

Round 1

Reviewer 1 Report

Comments and Suggestions for Authors

In their work, Liu et al investigated the possibility of uptake of the pro-drug PTOX in bacterial cells. A positive influence of the detergent CTAB was shown, which however also led to a reduced viability of the cells. As protection against CTAB, a putative zero-valent sulphur transporter, which has not yet been investigated in detail, was co-expressed in order to better scavenge the ROS generated by the addition of CTAB. All in all, the work is well written and the experiments performed confirm the statements made. However, I have some points to raise regarding the description of the methods and the logical structure of the results section that need to be addressed before publication.

Major points:

Table 1: The information given in the table are not sufficient to follow the plasmid features. There is no information on replication ori, except for pBBR1MCS2 the included resistance gene etc. As far as I know pBBR1MCS2 cannot be purchased from Invitrogen but Addgene. The reference “this lab” did not allow to identify and evaluate the used vectors. Thus if no reference can be provided because the plasmid is from a labstock at least a map should be provided in the supplements better the sequence (or link to the sequence) of a parent plasmid and a detailed description of the cloning procedures. Moreover the plasmids pBBR1MCS5, pCL120 and pTrc99a are missing in the table.

Line 171ff: The structure of the first paragraphs of the results section makes it difficult to follow the story that is to be shown. It is completely unclear why potential membrane transporters are first searched for and why an yet uncharacterised transporter is then chosen to investigate the effect of a detergent on the uptake of a substrate that is presumably not taken up via this transporter. Hence, the order of the first sections should be reconsidered. In a way that first a introducing sentence to the problem is given (PTOX modification is interesting/important but it is not taken up well by the bacteria). 2nd CTAB improves the PTOX uptake in BL21 (Fig1 left half) but higher concentrations are toxic (Fig 2A) most likely due to increased ROS. 3rd search for potential transporters for zero valent sulfur (current paragraph 3.1). 4th experiments using the YedE1E2 strain (Fig. 1 right half and Fig.2B/C.)

Figs. 2 and 4-7: Please consider to use colored images.

Fig.2 How long did the cells grow before CTAB addition. This should be included in the growth curves as this would help to see the inhibitory effect of CTAB.

Line 225: There is no real growth until 15h for E. coli (YedE1E2) with 600 µM CTAB. When comparing the growth rate of this strain during the last 5 hours of cultivation it seems to be the same as the growth rate of the strain without CTAB during the first 5 h of cultivation. A possible explanation that should be considered is the accumulation of mutations in the culture that allows to resist CTAB.

Line 395: The entire value of the research depends on the  feaseability of modification of the prodrug inside the bacterial cell. Is this already possible? What are the prerequisites? Where are the limitations? Is such a process economically feasible? These questions should be discussed critically.

Fig. 9: Why is there an out group protein (WP083113426.1) right in the middle of the tree. I think that out groups should only appear at the edge of the tree otherwise they cannot really considered as an out group. Moreover, the fond size is very small and can hardly been read (especially when printed) and when zooming in the figure is not sharp anymore. Hence, a version with higher resolution should be provided.

Minor points:

Line 21: Please consider to use “the … transporter” instead of “a … transporter”.

Line 22: Do you mean “presence” of CTAB.

Line 23: This sentence should be rephrased. If YedE1E2 is a transporter it should be stated as “is…a hypothetical transporter”. Following the assumption that YedE1E2 is a transporter it is unlikely that it reduces ROS itself but provides the prerequisite for ROS reduction by increasing the import of zero valent sulfur.

Line 55: The sentence “It accumulates in mitochondria…” is out of context. The paragraph describes the influence of CTAB on bacteria, however bacteria don’t possess mitochondria. Please rephrase.

Line 66: The space between “sulfide:” and “quinone” should be removed throughout the entire manuscript.

Line 82: What is meant with “E. coli containing a plasmid…usually compared with…”? Please explain and rephrase. Moreover, there is a verb missing in this sentence.

Line 85: Since there are different LB recipes please provide the composition or at least a reference for the used LB. Similarly, the composition of the MM medium has to be provided.

Line 89: Please provide a reference for the plasmids pBBR1MCS2, pBBR1MCS5, pCL1920 and pTrc99a.

Line 117: BL21 has to be in capital letters. Please check throughout the entire manuscript.

Line 121: How did you separate membrane proteins from membrane particles after resuspension in Tris buffer. Please comment.

Line 138: What is meant with “…its recombinant cells”? Please rephrase.

Line 143: The sentence “The cell suspension…” has to be rephrased.

Line 177: Please insert an “of” between “are upstream” and “the blh gene”.

Line 188: Please use “are small proteins” instead of “were small proteins”.

Line 198: The part “…the following CTAB experiments…” is slang and should be rephrased or removed.

Line 200: The toxicity of CTAB is not investigated in paragraph 3.2 but in 3.3. and Fig.2.

Line 213: either use “YedE1E2 production induced with IPTG” or “YedE1E2 production induced by addition of IPTG”

Line 347: “…which is CpYedE1 Cys104 and CpYedE2 Cys101, respectively”.

Line 379: Gram positive should be written out.

Line 396: “Integrity” instead of “Integrality”?

Line 436: The reference Wang et al occurs not as number.

Comments on the Quality of English Language

The English is mostly fine.

Author Response

Reviewer 1:Thank you for the valuable comments and suggestions. We are addressed them as indicated below.

Major points:

  1. Table 1: The information given in the table are not sufficient to follow the plasmid features. There is no information on replication ori, except for pBBR1MCS2 the included resistance gene etc. As far as I know pBBR1MCS2 cannot be purchased from Invitrogen but Addgene. The reference “this lab” did not allow to identify and evaluate the used vectors. Thus if no reference can be provided because the plasmid is from a labstock at least a map should be provided in the supplements better the sequence (or link to the sequence) of a parent plasmid and a detailed description of the cloning procedures. Moreover the plasmids pBBR1MCS5, pCL1920 and pTrc99a are missing in the table.

Done. References are provided, the construction procedures are explained briefly in lines 100-105. The construction of the plasmid pCL1920 with target genes are illustrated in the supplements. pBBR1MCS5 was not used and was removed from the manuscript.

  1. Line 171ff: The structure of the first paragraphs of the results section makes it difficult to follow the story that is to be shown. It is completely unclear why potential membrane transporters are first searched for and why an yet uncharacterised transporter is then chosen to investigate the effect of a detergent on the uptake of a substrate that is presumably not taken up via this transporter. Hence, the order of the first sections should be reconsidered. In a way that first a introducing sentence to the problem is given (PTOX modification is interesting/important but it is not taken up well by the bacteria). 2nd CTAB improves the PTOX uptake in BL21 (Fig1 left half) but higher concentrations are toxic (Fig 2A) most likely due to increased ROS. 3rd search for potential transporters for zero valent sulfur (current paragraph 3.1). 4th experiments using the YedE1E2 strain (Fig. 1 right half and Fig.2B/C.)

Yes. The logic of the transition between YedE1E2 and CTAP/PTOX was not clear in the results section. We added several sentences to smooth the transition (Lines 203-207).

  1. 2 and 4-7: Please consider to use colored images.

Done.

  1. 2 How long did the cells grow before CTAB addition. This should be included in the growth curves as this would help to see the inhibitory effect of CTAB.

 Done. Information added in the lengend.

  1. Line 225: There is no real growth until 15h for  coli(YedE1E2) with 600 µM CTAB. When comparing the growth rate of this strain during the last 5 hours of cultivation it seems to be the same as the growth rate of the strain without CTAB during the first 5 h of cultivation. A possible explanation that should be considered is the accumulation of mutations in the culture that allows to resist CTAB.

Response: We added the following: “When both E. coli BL21 and E. coli (YedE1E2) started to grow in the presence of CTAB, the growth rates were similar to those without CTAB. The delayed growth with CTAB could be due to the accumulation of mutations or the activation of resistant genes, which could be further investigated.” (Lines 235-238)

  1. Line 395: The entire value of the research depends on the feasibility of modification of the prodrug inside the bacterial cell. Is this already possible? What are the prerequisites? Where are the limitations? Is such a process economically feasible? These questions should be discussed critically.

Done, Lines 407-410.

“Biosynthesis of complex eukaryotic metabolites or derivatives in prokaryotic organisms is an ongoing trend in the drug industry [39-42]. Compared to chemical synthesis which may be cumbersome with lots of protection and deprotection steps, microbial transformation could be a good choice to modify the structure of podophyllotoxin due to the strict stereo- and regioselectivity, mild production conditions and easy operations [43,44].”

  1. 9: Why is there an out group protein (WP083113426.1) right in the middle of the tree. I think that out groups should only appear at the edge of the tree otherwise they cannot really considered as an out group. Moreover, the fond size is very small and can hardly been read (especially when printed) and when zooming in the figure is not sharp anymore. Hence, a version with higher resolution should be provided.

 Done. The out group is now properly located at one end (Fig. 9).

Minor points:

  1. Line 21: Please consider to use “the … transporter” instead of “a … transporter”.

Done, line 21.

  1. Line 22: Do you mean “presence” of CTAB.

Done, line 22.

  1. Line 23: This sentence should be rephrased. If YedE1E2 is a transporter it should be stated as “is…a hypothetical transporter”. Following the assumption that YedE1E2 is a transporter it is unlikely that it reduces ROS itself but provides the prerequisite for ROS reduction by increasing the import of zero valent sulfur.

Done, line 23-25.

  1. Line 55: The sentence “It accumulates in mitochondria…” is out of context. The paragraph describes the influence of CTAB on bacteria, however bacteria don’t possess mitochondria. Please rephrase.

Done. The sentence is not essential, and it is removed.

  1. Line 66: The space between “sulfide:” and “quinone” should be removed throughout the entire manuscript.

Done.

  1. Line 82: What is meant with “ colicontaining a plasmid…usually compared with…”? Please explain and rephrase. Moreover, there is a verb missing in this sentence.

Done, lines 83-85.

  1. Line 85: Since there are different LB recipes please provide the composition or at least a reference for the used LB. Similarly, the composition of the MM medium has to be provided.

Done. Lines 85-86.

  1. Line 89: Please provide a reference for the plasmids pBBR1MCS2, pBBR1MCS5, pCL1920 and pTrc99a.

Done, line 90-91. pBBR1MCS5 was not used, and it was removed.

  1. Line 117: BL21 has to be in capital letters. Please check throughout the entire manuscript.

Done, Line 123.

  1. Line 121: How did you separate membrane proteins from membrane particles after resuspension in Tris buffer. Please comment.

We didn’t separate membrane proteins from membrane particles. We resuspended the membrane pellets by stirring with a glass rod to make the particles suspended in the buffer, and the suspension was used for the analysis.

  1. Line 138: What is meant with “…its recombinant cells”? Please rephrase.

Done, line 141.

  1. Line 143: The sentence “The cell suspension…” has to be rephrased.

Done, line 146.

  1. Line 177: Please insert an “of” between “are upstream” and “the blh gene”.

Done, line 181.

  1. Line 188: Please use “are small proteins” instead of “were small proteins”.

Done, line 192.

  1. Line 198: The part “…the following CTAB experiments…” is slang and should be rephrased or removed.

Done, the sentence is removed.

  1. Line 200: The toxicity of CTAB is not investigated in paragraph 3.2 but in 3.3. and Fig.2.

Done. The topic sentence of the paragraph was changed.

  1. Line 213: either use “YedE1E2 production induced with IPTG” or “YedE1E2 production induced by addition of IPTG”

Done, line 217.

  1. Line 347: “…which is CpYedE1 Cys104 and CpYedE2 Cys101, respectively”.

Done, line 353.

  1. Line 379: Gram positive should be written out.

Done, line 384.

  1. Line 396: “Integrity” instead of “Integrality”?

Done, line 407.

  1. Line 436: The reference Wang et al occurs not as number.

Done, line 448.

Reviewer 2 Report

Comments and Suggestions for Authors

This is an interesting and detailed work that is logical in terms of experiments, results and explanation.

There is an issue concerning very basic terminology in the redox field. The authors refer to GSSH as oxidized glutathione (GSH). GSSH is GSH persulfide. This is a species with an active thiolate thus short lived and most likely will end up as GSSG to be reduced by GSH reductase back to GSH. Therefore, I suggest that a more proper term for oxidized GSH in the context of its cellular status could be GSSG (and not GSSH).

It would be nice if the authors could suggest a mechanism for the transport of sulfur by the transporter, it would add more value to their work.

Line 85 Cupriavidus pinatubonensis: introduce here the abbreviation C. pinatubonensis.

Author Response

Reviewer 2: Thank you for the valuable comments and suggestions. We are addressed them as indicated below.

  1. There is an issue concerning very basic terminology in the redox field. The authors refer to GSSH as oxidized glutathione (GSH). GSSH is GSH persulfide. This is a species with an active thiolate thus short lived and most likely will end up as GSSG to be reduced by GSH reductase back to GSH. Therefore, I suggest that a more proper term for oxidized GSH in the context of its cellular status could be GSSG (and not GSSH).

Yes. The oxidized GSH is GSSG. GSSH is glutathione persulfide, and it has dual functions. It can react with H2O2, and it also induces the formation of GSSG. We added a sentence to make the point (Lines 69-70).

  1. It would be nice if the authors could suggest a mechanism for the transport of sulfur by the transporter, it would add more value to their work.

The mechanism is currently unclear. However, we believe that the formation of GSSH or other cellular persulfides drives the uptake. The information is suggested in the conclusion section (newly added, lines 465-471).

  1. Line 85 Cupriavidus pinatubonensis: introduce here the abbreviation  pinatubonensis.

Done. Line 86.

Round 2

Reviewer 1 Report

Comments and Suggestions for Authors

In the revised version of their manuscript the authors followed most of my suggestions, thus significantly improving the paper. However, some of my comments have not been addressed satisfactorily so far.

Major comments:

Line 177ff: The authors tried to improve the structure of the results section, however, the newly added sentences did not help. The experiments that lead to the conclusion that CTAB enhances PTOX uptake (line 204) are first conducted in paragraph 3.2; the toxicity of CTAB is investigated in paragraph 3.3 and the CTAB resistance is shown in Fig. 1. Consequently, I still cannot see the logic in the story that is to be shown.

I suggest to start the results section with paragraph 3.2 (including an introducing sentence stating the importance of PTOX uptake in bacteria) followed by the current paragraph 3.1 (including an introducing sentence which explains how the authors came to the idea of investigating a potential transporter)

Fig. 2: The authors stated that they induced the cells when they reached OD 0.6. In my first review I ask to include the information how long the cells grow until they reached this OD and to include this information in the figure and the figure legend.

LIne 405ff: The authors tried to responde to my previous comment, however, I think there is still room for improvement of the discussion. Please include some information regarding the limitations and prerequisites (e.g. co-expression of enzymes) as well as the economic feasability of such a process.

Minor comment:

Line 234: Not the growth delay is due to accumulation of mutations but the start of the growth after 15 h might be the result of the accumulation of mutations. This should be clarified.

Author Response

Reviewer I round 2. Thank you for the valuable suggestions. Responses are given below.

In the revised version of their manuscript the authors followed most of my suggestions, thus significantly improving the paper. However, some of my comments have not been addressed satisfactorily so far.

Major comments:

Line 177ff: The authors tried to improve the structure of the results section, however, the newly added sentences did not help. The experiments that lead to the conclusion that CTAB enhances PTOX uptake (line 204) are first conducted in paragraph 3.2; the toxicity of CTAB is investigated in paragraph 3.3 and the CTAB resistance is shown in Fig. 1. Consequently, I still cannot see the logic in the story that is to be shown.

I suggest to start the results section with paragraph 3.2 (including an introducing sentence stating the importance of PTOX uptake in bacteria) followed by the current paragraph 3.1 (including an introducing sentence which explains how the authors came to the idea of investigating a potential transporter)

Response: Thank you. The two sections are rearranged, and the test of E. coli (YedE1E2) is explained. Good suggestion.

Fig. 2: The authors stated that they induced the cells when they reached OD 0.6. In my first review I ask to include the information how long the cells grow until they reached this OD and to include this information in the figure and the figure legend.

Response: The information is added.

LIne 405ff: The authors tried to responde to my previous comment, however, I think there is still room for improvement of the discussion. Please include some information regarding the limitations and prerequisites (e.g. co-expression of enzymes) as well as the economic feasability of such a process.

Response: The paragraph was revised, and the limitations are discussed in the last sentence of the paragraph (Lines 425-428).

“Further investigations are necessary to demonstrate whether YedE1E2 can be used to promote CTAB-enhanced uptake of prodrugs like PTOX for the production of their derivatives in recombinant microorganisms and whether the production is economically feasible.”

Minor comment:

Line 234: Not the growth delay is due to accumulation of mutations but the start of the growth after 15 h might be the result of the accumulation of mutations. This should be clarified.

Clarified.

Round 3

Reviewer 1 Report

Comments and Suggestions for Authors

The authors have improved the manuscript according to my suggestions. I think the manuscript can be accepted now eventhough I would add an arrow in Fig. 2 indicating induction and an introducing sentence to paragraf 3.1.

Author Response

Arrows were added in Fig. 2A&B. An introducing sentence to Paragraph 3.1 was added.